# A Minimalist Approach to LLM Reasoning: from Rejection Sampling to Reinforce

## Abstract

Reinforcement learning (RL) methods such as Group Relative Policy Optimization (GRPO) have recently emerged as a leading approach for enhancing the reasoning ability of large language models (LLMs). Yet, the precise sources of their effectiveness remain unclear. In this work, we systematically decompose GRPO by benchmarking it against simpler REINFORCE-style baselines to identify its core components. Our analysis reveals a clear hierarchy: (i) iterative, online data collection is the dominant driver of performance, enabling even simple positive-only fine-tuning (e.g., RAFT) to be surprisingly strong; (ii) negative signals primarily sustain exploration by preventing rapid entropy collapse; and (iii) GRPO's main benefit stems not from reward normalization itself, but from the **implicit data filtering effect** it induces by discarding prompts with uniform rewards (all-correct or all-incorrect). Guided by this insight, we propose REINFORCE-REJ, a minimal variant that makes filtering explicit. REINFORCE-Rej matches GRPO's performance while being simpler and more KL-efficient. These findings suggest that principled data filtering, rather than algorithmic complexity, is the key to robust RL for LLMs.

## 1 Introduction

Group Relative Policy Optimization (GRPO) (Shao et al., 2024) has emerged as one of the most widely adopted reinforcement learning algorithms for fine-tuning Large Language Models (LLMs), most notably powering the DeepSeek-R1 model (DeepSeek-AI et al., 2025). As a REINFORCE-style method (Williams & Peng, 1991), GRPO offers a lightweight alternative to Proximal Policy Optimization (PPO) (Schulman et al., 2017), circumventing the need for a critic network and the instability often associated with value-function estimation. Its simplicity, scalability, and strong empirical performance have made GRPO an attractive choice for reinforcement learning with verifiable rewards (RLVR), where responses can be checked against ground truth labels or rules. Despite this rapid adoption, however, the fundamental mechanisms underlying GRPO's effectiveness remain opaque. Is its strength attributable to the group-based reward normalization scheme, or are there deeper, more generalizable principles at play? Clarifying this distinction is crucial, not only for interpreting prior successes but also for guiding the design of the next generation of reinforcement learning algorithms for reasoning LLMs.

In this work, we revisit GRPO with the goal of disentangling its essential ingredients. To do so, we systematically compare it against two natural and simpler baselines: *Rejection-Sampling Fine-Tuning (RAFT)* (Dong et al., 2023), which fine-tunes exclusively on positively rewarded samples, and the classical vanilla REINFORCE. By decomposing GRPO and conducting targeted ablation studies, we arrive at several key insights:

1. **Online exploration is the decisive factor.** RAFT, though conceptually and algorithmically simple, turns out to be a surprisingly competitive baseline: it converges quickly and achieves within 3% of GRPO's performance. Yet, its reliance on positive-only updates inevitably causes rapid entropy collapse, stalling exploration and capping performance. GRPO avoids this failure mode by incorporating negative updates, which play a critical role in sustaining exploration and maintaining policy diversity.

2. **GRPO's performance edge comes from implicit filtering, not group normalization.** While GRPO is often framed around its group-wise normalization trick, our experiments reveal that normalization itself has negligible effect. Instead, the true advantage arises from an *implicit filtering effect*: training batches in which all responses receive identical rewards (all-correct or all-incorrect) contribute zero gradient signal and are effectively discarded. This implicit filter prevents the policy from being pushed in uninformative or destabilizing directions.

3. **Explicit filtering yields a simpler yet equally strong baseline.** Inspired by this observation, we introduce REINFORCE-REJ, a minimalist variant of REINFORCE equipped with an explicit rejection filter that discards uniform-reward prompts. Despite its simplicity, REINFORCE-Rej matches GRPO's performance while being more KL-efficient and easier to analyze, highlighting that stable RLVR may not require group normalization at all.

Collectively, these findings suggest that in current RLVR practice, *careful data filtering may matter more than sophisticated algorithmic machinery*. The instability induced by "extreme prompts"—for example, cases where all responses are incorrect and the policy is driven solely by an "unlearning" signal—can derail training. Simple yet principled filtering of such cases, even at the cost of a small statistical bias, leads to a more stable and effective learning process. Beyond clarifying the sources of GRPO's success, our results point to a broader design principle: when working with verifiable rewards, the choice of what data to *ignore* can be just as important as how to update from the data that remains. We hope these insights not only deepen the understanding of GRPO and its relatives but also inspire the development of even more efficient, robust, and theoretically grounded reinforcement learning algorithms for reasoning-capable LLMs.

## 2 From Rejection Sampling to Relative Policy Optimization

We frame the task of improving a language model, or policy $\pi_\theta$, as maximizing the expected reward on a given prompt distribution. Concretely, given a prompt $x$ drawn from a distribution $d_0$, the policy generates a response $a \sim \pi_\theta(\cdot|x)$. We assume access to a verifier $r^\star(x, a) \in \{0, 1\}$ that assigns a binary reward of 1 for a correct response and 0 otherwise. The goal is to optimize the policy parameters $\theta$ such that the expected reward is maximized:

$$J(\theta) = \mathbb{E}_{x \sim d_0, a \sim \pi_\theta(\cdot|x)}[r^\star(x, a)]. \tag{1}$$

This setup, which we refer to as reinforcement learning with verifiable rewards (RLVR), admits multiple algorithmic strategies that trade off simplicity, stability, and efficiency. In this section, we present three closely related families of methods as a natural progression: starting from Rejection-Sampling Fine-Tuning (RAFT), which relies on positive-only supervised updates; extending to vanilla REINFORCE, which incorporates unbiased stochastic gradients of $J(\theta)$; and culminating in Group Relative Policy Optimization (GRPO), which stabilizes REINFORCE through relative normalization and implicit filtering. By laying them out in this sequence, we highlight their shared structure and clarify how each step incrementally addresses the shortcomings of the previous.

### 2.1 RAFT: Rejection Sampling Fine-Tuning

We start with a simple yet powerful baseline *Rejection-Sampling Fine-Tuning (RAFT)* (Dong et al., 2023). RAFT operates through an iterative process:

1. **Generation**: For each prompt in a batch, sample $n$ responses using the current policy $\pi_{\theta_t}$.

2. **Filtering**: Create a dataset $\mathcal{D}$ containing only the prompt-response pairs $(x, a)$ that received a positive reward, i.e., $r^\star(x, a) = 1$.

3. **Update**: Fine-tune the policy on this filtered dataset using a standard supervised learning objective (maximum likelihood estimation):

$$\mathcal{L}_{\text{RAFT}}(\theta) = -\mathbb{E}_{(x,a) \in \mathcal{D}}[\log \pi_\theta(a|x)]. \tag{2}$$

The initial intuition behind RAFT is to iteratively imitate an increasingly strong "best-of-n" policy by filtering out the low-quality data. But we will see it can also be viewed as a special case of policy gradient.

## 2.2 REINFORCE: A General Policy Gradient Framework

The objective in Equation (1) can be directly optimized using policy gradient methods. The REINFORCE algorithm (Williams & Peng, 1991) computes the gradient of the objective as:

$$\nabla_\theta J(\theta) = \mathbb{E}_{x \sim d_0, a \sim \pi_\theta(\cdot|x)}[r^\star(x,a) \cdot \nabla_\theta \log \pi_\theta(a|x)].$$

In practice, we use the following stochastic gradient estimator:

$$g_\theta(x,a) = (r^\star(x,a) - b(x)) \cdot \nabla_\theta \log \pi_\theta(a|x),$$

where $x \sim d_0, a \sim \pi_{\theta_t}(\cdot|x)$. This provides a clear connection to RAFT. When using a binary reward $r^\star \in \{0,1\}$, the gradient term $r^\star(x,a) \cdot \nabla_\theta \log \pi_\theta(a|x)$ is non-zero only for successful trajectories where $r^\star = 1$. In this scenario, the policy gradient update recovers the RAFT-like algorithms (Equation (2)), which **only learn from positive samples**.

While RAFT learns exclusively from successes, policy gradient methods can also learn from failures. The most direct way to do this is to introduce an explicit negative signal. For instance, the vanilla REINFORCE algorithm can be implemented by mapping the binary outcomes to a reward set like $\{-1, 1\}$, which encourages good responses and penalizes bad ones.

## 2.3 GRPO: Introducing a Relative Advantage

To incorporate learning from negative samples, we can also introduce a baseline $b(x)$ to the reward term, which reduces variance without changing the expected gradient. The policy gradient estimator then becomes:

$$g_\theta(x,a) = (r^\star(x,a) - b(x)) \cdot \nabla_\theta \log \pi_\theta(a|x).$$

Group Relative Policy Optimization (GRPO) (Shao et al., 2024) extends the baseline concept by calculating a prompt-specific, or "group-relative" advantage. For each prompt $x$, GRPO samples a group of $n > 1$ responses $\{a_1, \ldots, a_n\}$ and their corresponding rewards $\{r_1, \ldots, r_n\}$. The advantage for the $i$-th response is then calculated by normalizing its reward using the group's mean and standard deviation:

$$A_{\text{GRPO}}(x, a_i) = \frac{r_i - \bar{r}}{\sigma_r + \eta}, \tag{3}$$

where $\bar{r} = \frac{1}{n}\sum_{j=1}^n r_j$, $\sigma_r = \sqrt{\frac{1}{n}\sum_{j=1}^n (r_j - \bar{r})^2}$, and $\eta$ is a small constant for numerical stability. Thus, GRPO can be seen as a REINFORCE variant that employs a dynamically computed baseline (the group mean) and an adaptive scaling factor (the group standard deviation).

Table 1: Comparison of the core learning signals for the three algorithms studied. All are implemented within the same PPO-style framework provided in Equation (4).

| Algorithm | Advantage Calculation $A(x,a)$ | Key Characteristic |
|---|---|---|
| RAFT++ | $A(x,a) = r^\star(x,a) \in \{0,1\}$ | Learning from positive samples only. |
| REINFORCE | $A(x,a) = r(x,a) \in \{-1,1\}$ | Introduces negative signals via reward mapping. |
| GRPO | $A(x,a) = (r_i - \bar{r})/(\sigma_r + \eta)$ | Introduces negative signals via baseline and reward shaping. |

## 2.4 PPO-like enhancement: importance sampling and clipping

A major limitation of pure on-policy algorithms like REINFORCE is sample inefficiency, as each batch of data can only be used for a single gradient update. Following modern best practices (Schulman et al., 2017;

Shao et al., 2024), in practice, we usually implement these algorithms with two techniques from PPO to improve stability and efficiency.

First, we use **importance sampling** to allow for multiple updates on the same batch of data. We collect data with an old policy $\pi_{\theta_{\text{old}}}$ and update the current policy $\pi_\theta$ using a ratio $\rho_t(\theta) = \frac{\pi_\theta(a|x)}{\pi_{\theta_{\text{old}}}(a|x)}$. Second, to prevent destructively large updates when $\pi_\theta$ diverges from $\pi_{\theta_{\text{old}}}$, we use **clipping**. This leads to the following general objective function for a given advantage function $A(x, a)$:

$$\mathcal{L}_{\text{PG}}(\theta) = -\mathbb{E}_{(x,a) \sim \pi_{\theta_{\text{old}}}} \left[ \min \left( \rho_t(\theta) A(x, a), \text{clip}(\rho_t(\theta), 1 - \epsilon, 1 + \epsilon) A(x, a) \right) \right]. \tag{4}$$

The specific algorithm—REINFORCE or GRPO—is determined solely by the choice of the advantage function $A(x, a)$. For LLMs, this loss is typically applied at the token level to provide more granular learning signals. All policy gradient methods in our experiments are implemented using this PPO-style objective, with a summarization in Table 1. While these enhancements are quite standard in practice, we also include an experiment in Appendix A.2 to verify their effectiveness.

## 3 Experiments: Decomposing Group Relative Policy Optimization

### 3.1 Experiment Setup

We focus on the mathematical reasoning task in this project. The implementations are mainly based on the verl (Sheng et al., 2024) framework.

**Dataset and Models.** We train the models using the prompt set Numina-Math (Beeching et al., 2024), which consists of approximately 860k math problems and labeled ground-truth answers. The sources of Numina-Math ranges from Chinese high school math exercises to US and international mathematics olympiad competition problems. We conduct experiments with both Qwen2.5-Math-7B-base, and LLaMA-3.2-3B-instruct for generality. We use the default chat template of these models and use CoT prompting: "Let's think step by step and output the final answer within \boxed{}".

**Hyper-parameters.** We follow most of the hyper-parameter setups recommended in the verl framework for the Reinforce, GRPO, and PPO training. The hyper-parameters for RAFT and RAFT++ are also the same with the GRPO script. Specifically, we use the AdamW optimizer with a learning rate of $1 \times 10^{-6}$. We sample 1024 prompts per iteration, and generate $n = 4$ responses per prompt for RAFT and GRPO. The training mini-batch size is set to be 512. The models are allowed to generate 4096 tokens at most during training. More detailed scripts are available in the GitHub repository. For the baseline of iterative DPO, we use the codebase developed in Zhang et al. (2025).

**Evaluation.** We evaluate the models' reasoning ability by Math500 (Hendrycks et al., 2021), Minerva Math (Lewkowycz et al., 2022), Olympiad Bench (He et al., 2024). We do not include the popular AIME2024 benchmark since it only consists of 30 problems. In our preliminary experiments, we observe that the trend on this benchmark is very noisy for all the considered algorithms. We mainly use average@16 to evaluate our models, where we generate 16 responses per prompt with temperature 1.0, and use the average accuracy as the metric. The models are allowed to generate 4096 token at most.

### 3.2 RAFT++ vs. GRPO: Training Dynamics of Positive-Only vs. Full RL Methods

**Online data collection is the primary driver of performance.** Our first key finding, presented in Table 2, is that the simple RAFT++ algorithm achieves remarkably strong performance. On the Qwen2.5-Math-7B model, RAFT++ reaches an average accuracy of 54.9, trailing the more complex GRPO (56.3) by only 1.4 points. Similarly, for LLaMA-3.2-3B-instruct model, RAFT++ enjoys an average accuracy of 26.6, which is also very close to the 28.4 of GRPO.

To disentangle these factors, we conduct an additional experiment shown in Figure 1. We compare four methods: (1) **Offline RS**, which fine-tunes on a large dataset collected only once from the base model;

| Model | Algorithm | Math500 | Minerva Math | Olympiad Bench | Average |
|---|---|---|---|---|---|
| | Base | 41.3 | 11.0 | 18.6 | 23.6 |
| | Offline RS Fine-tuning | 70.0 | 26.4 | 32.9 | 43.2 |
| Qwen2.5-Math-7B-base | RAFT++ | 80.1 | 44.4 | 40.3 | 54.9 |
| | RAFT++ with Clip Higher | 80.2 | 44.9 | 43.3 | 56.1 |
| | GRPO | 81.3 | 45.5 | 42.2 | 56.3 |
| | Base | 26.3 | 7.4 | 5.5 | 13.1 |
| LLaMA-3.2-3B-instruct | RAFT++ | 47.2 | 17.3 | 15.2 | 26.6 |
| | RAFT++ with Clip Higher | 47.4 | 19.1 | 16.3 | 27.6 |
| | GRPO | 49.2 | 19.3 | 16.8 | 28.4 |

Table 2: Performance of different algorithms across three benchmarks including Math500 (Hendrycks et al., 2021), Minerva Math (Lewkowycz et al., 2022), and Olympiad Bench (He et al., 2024). We report average@16 accuracy with a temperature 1.0 and a maximal generation length of 4096 tokens.

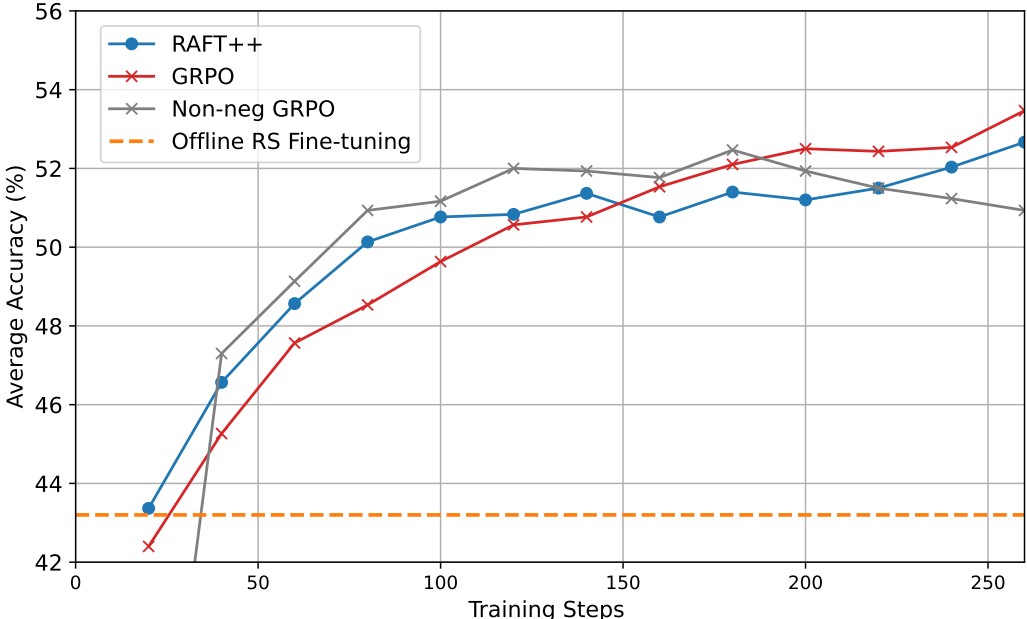

Figure 1: The test reward curves of RAFT++, GRPO, and Non-neg GRPO where mask the negative gradient during the GRPO training. The offline RS Fine-tuning uses the base model to collect data on 20K prompts using best-of-16 policy without further online exploration.

(2) **RAFT++**, the standard online rejection sampling fine-tuning method; (3) **Non-negative GRPO**, a variant where we mask gradients from negative samples, making it an online, positive-only method like RAFT++; and (4) standard **GRPO**. The Offline RS baseline saturates almost immediately, demonstrating that without an updating policy for data generation, performance gains are severely limited. In contrast, all three online methods significantly outperform it. This shows that the **online loop—iteratively generating samples from an increasingly capable policy—is the most important factor for achieving strong performance**. Furthermore, while Non-negative GRPO and RAFT++ are competitive, the full GRPO algorithm achieves the best final score, indicating that negative signals provide an additional, valuable boost on top of the gains from online learning.

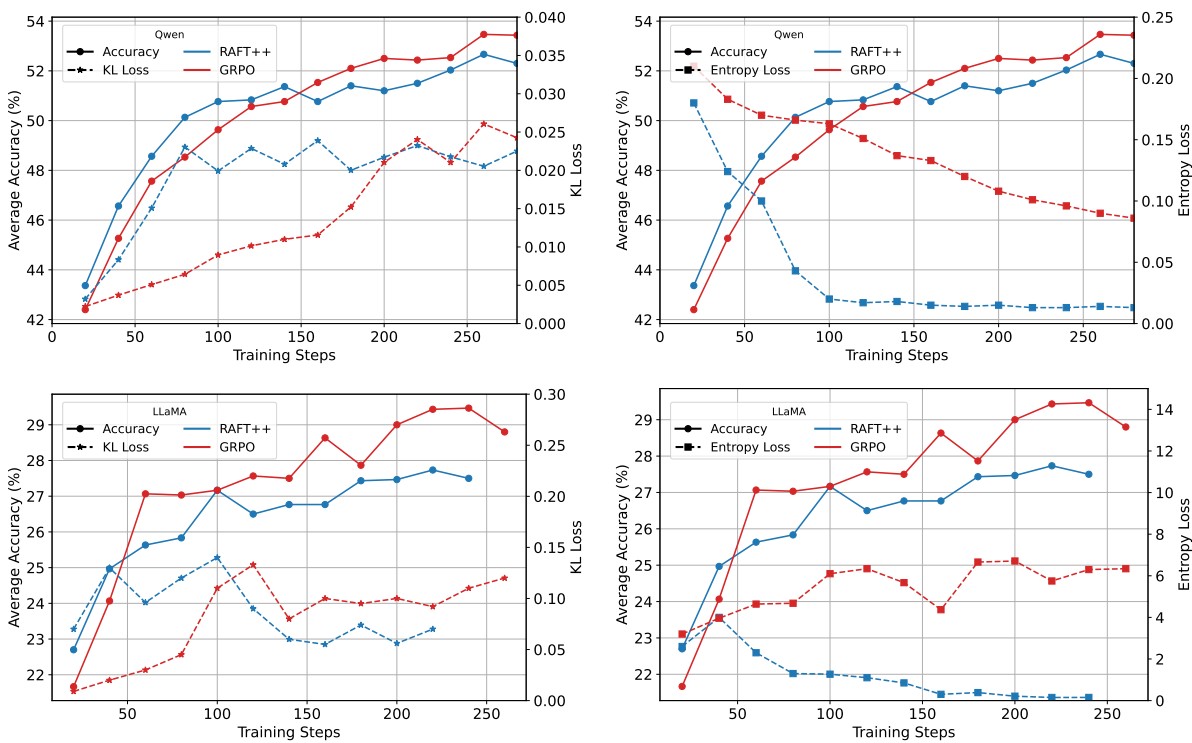

Figure 2: Training dynamics of RAFT++ and GRPO on Qwen2.5-Math-7B (top row) and LLaMA-3.2-3B (bottom row). We plot the test reward (left) and policy entropy (right) against training iterations. We also plot the KL loss in the left column and the policy entropy loss in the right column.

**RAFT++ achieves faster early-stage convergence but is surpassed by GRPO in later training.** To understand what accounts for this small yet persistent gap, we examine the training dynamics of both algorithms in Figure 2. We observe that RAFT++ exhibits much faster convergence in the early training stages, quickly surpassing GRPO. However, its progress noticeably slows and begins to plateau around iteration 100. In contrast, GRPO learns more steadily and continues to improve, ultimately overtaking RAFT++ in final performance.

**The plateau is linked to a rapid collapse in policy entropy.** The reason for RAFT++'s plateau becomes evident when we analyze its policy entropy (Figure 2, right column). RAFT++, which learns exclusively from positive samples, aggressively exploits successful trajectories. This leads to a rapid collapse in policy entropy, as the model's output distribution quickly becomes less diverse. Crucially, the performance of RAFT++ begins to stagnate precisely when its entropy bottoms out. We hypothesize that this loss of diversity hurts exploration, preventing the model from discovering new and potentially better reasoning paths. GRPO, by learning from both positive and negative signals, maintains a higher policy entropy for longer, facilitating sustained exploration and continuous improvement.

**Mitigating entropy collapse in RAFT++ narrows the gap with GRPO.** To directly test our hypothesis, we conducted an experiment to see if preserving RAFT++'s entropy would allow it to continue learning. We incorporate the "clip higher" technique from Yu et al. (2025b), which uses a larger upper clipping boundary for the probability ratio. As shown in Figure 3, this simple modification helps RAFT++ maintain a higher entropy level throughout training. The performance impact, detailed in Table 2, is also significant. The enhanced RAFT++ variant substantially narrows the performance gap with GRPO, improving its average accuracy from 54.9 to 56.1 on Qwen2.5-Math-7B, bringing it much closer to GRPO's score of 56.3. This result provides strong evidence that a key advantage of GRPO is its ability to promote sustained exploration by preventing premature policy collapse.

| Model | Algorithm | Math500 | Minerva Math | Olympiad Bench | Average |
|---|---|---|---|---|---|
| | Base | 41.3 | 11.0 | 18.6 | 23.6 |
| Qwen2.5-Math-7B-base | REINFORCE | 80.1 | 40.7 | 40.9 | 53.9 |
| | GRPO | 81.3 | 45.5 | 42.2 | 56.3 |
| | REINFORCE-Rej | 81.9 | 44.2 | 43.1 | 56.4 |
| | Base | 26.3 | 7.4 | 5.5 | 13.1 |
| LLaMA-3.2-3B-instruct | REINFORCE | 45.9 | 13.7 | 13.0 | 24.2 |
| | GRPO | 49.2 | 19.3 | 16.8 | 28.4 |
| | REINFORCE-Rej | 50.1 | 19.3 | 16.1 | 28.5 |

Table 3: Performance comparison of REINFORCE, GRPO, and a simplified algorithm, REINFORCE-Rej. We report average@16 accuracy with a temperature 1.0 and a maximal generation length of 4096 tokens[2].

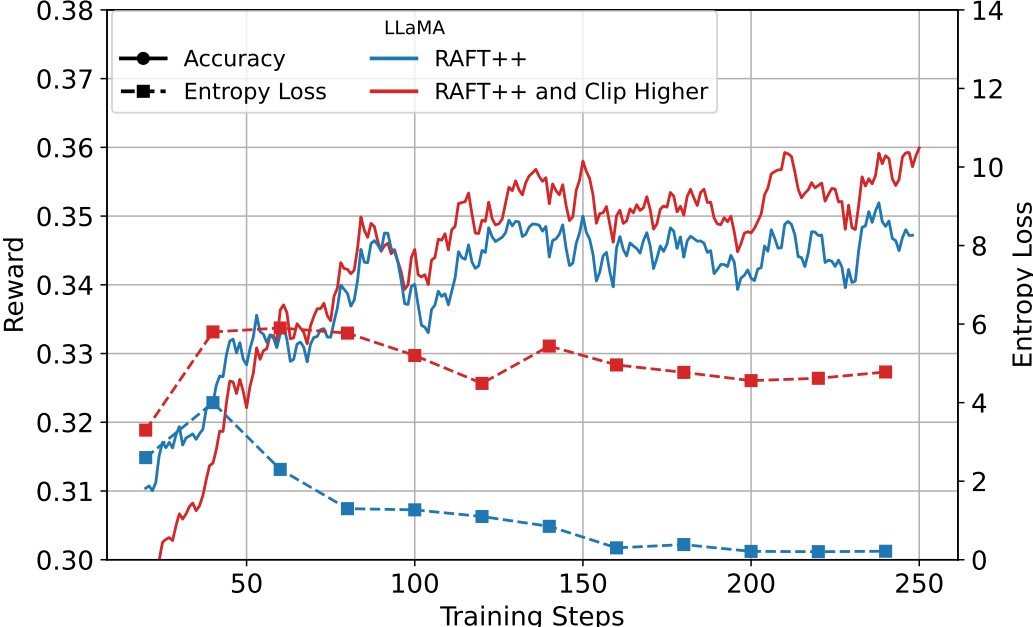

Figure 3: The training reward curves of RAFT++ and RAFT++ enhanced by clip higher trick, initialized from LLaMA-3.2-3B-instruct. We apply a moving average with a window size of 20 to smooth the curves.

## 3.3 REINFORCE vs. GRPO: The Role of Implicit Data Filtering

In the previous section, we established that online learning is critical and that negative signals are valuable for sustaining exploration. This leads to the next logical question: what is the most stable and effective way to incorporate these negative signals? The simplest method is vanilla REINFORCE, which maps rewards to $\{1, -1\}$. However, as shown in Table 3, vanilla REINFORCE significantly underperforms GRPO. This suggests that *how* negative signals are used is as important as using them at all.

**From REINFORCE to GRPO: what is the key role to the success of GRPO?** The primary differences between GRPO and vanilla REINFORCE lie in two aspects: the way of introducing negative signals and the application of reward normalization. To isolate the contributions of each component and

---

[2]The REINFORCE-Rej model was trained later in our research on an updated hardware environment. For a fair comparison, all other models reported in this table were re-run on this final setup. While this may cause minor numerical deviations from the learning curve figures we discussed, we confirmed that the overall learning dynamics and relative performance trends are consistent.

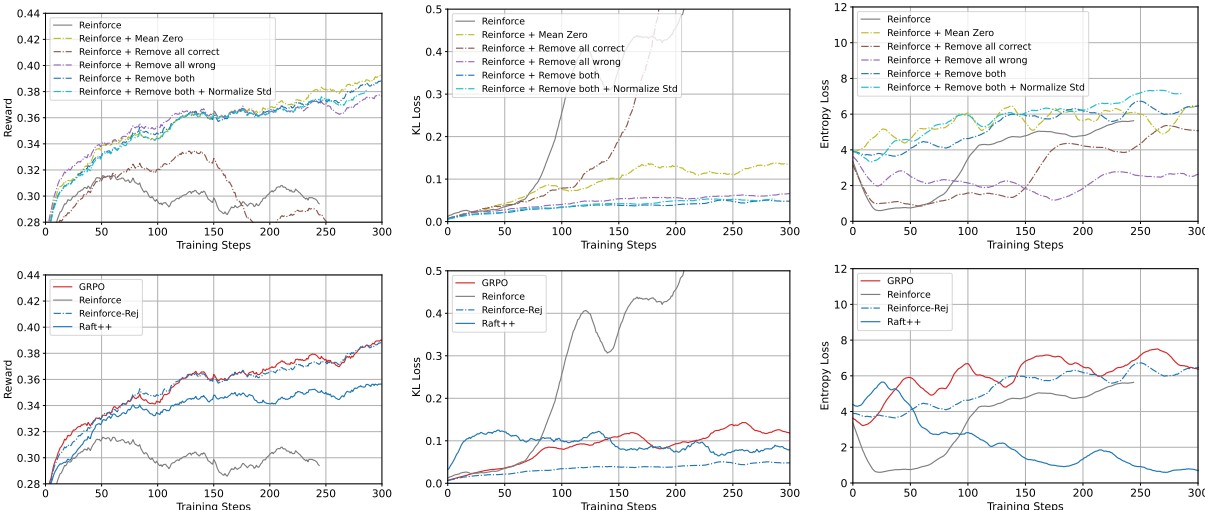

Figure 4: Ablation study on the components of GRPO and REINFORCE-type algorithms with LLaMA-3.2-3B-instruct. We compare GRPO with other REINFORCE-based variants to isolate the effects of removing incorrect samples, correct samples, and applying normalization. Removing incorrect samples ("Remove all wrong") provides the largest gain in reward, highlighting their harmful impact. In contrast, the reward of removing correct samples is still not satisfactory. Mean-zero normalization increases KL loss and destabilizes training. Normalizing by standard deviation shows minimal additional benefit. The variant "REINFORCE + Remove both" achieves a good balance between reward, KL stability, and entropy regularization. We transform the original reward using $(1 + r)/2$ so that the resulting value corresponds to the accuracy on the training data. We also apply a moving average with a window size of 20 to smooth the curves.

better understand their respective effects, we designed a set of controlled experiments to systematically evaluate their impacts. Specifically, we consider the following algorithms:

1. Reinforce: the vanilla one by mapping the reward to $\{-1, 1\}$;

2. REINFORCE + Mean Zero: we subtract the mean reward within each prompt;

3. REINFORCE + Remove all correct: we filter out prompts whose responses are entirely correct;

4. REINFORCE + Remove all wrong: we filter out prompts whose responses are entirely wrong;

5. REINFORCE + Remove both: remove both fully correct and fully incorrect prompts;

6. REINFORCE + Remove both + Normalized Std: in addition to removing both fully correct and fully incorrect prompts, we further divide the reward by its standard deviation within each prompt for normalization.

**Filtering uniformly incorrect samples is the most critical factor.** As shown in Figure 4, our ablation study reveals that the most significant performance improvement comes from sample filtering. The "REINFORCE + Remove all wrong" variant dramatically outperforms vanilla REINFORCE, clearly indicating that forcing the model to unlearn from prompts where it consistently fails is particularly harmful. This is likely because such samples provide high-variance, misleading gradients that can destabilize training. In contrast, removing only all-correct prompts yields little benefit. Combining both ("REINFORCE + Remove both") results in the most stable training and slightly better rewards, suggesting that filtering these extreme, low-signal prompts helps maintain effective exploration.

**Reward normalization itself offers little to no benefit.** Conversely, when isolated from their filtering effects, the normalization components of GRPO are less ineffective. We observe that applying mean-subtraction alone ("REINFORCE + Mean Zero") is very similar to the "REINFORCE + Remove both" but with worse KL loss. Moreover, applying standard deviation normalization on the top of data filtering ("REINFORCE + Remove both + Normalize Std") yields no additional gain over simply removing bad samples, suggesting that variance normalization is not a key contributor to performance.

Taken together, these results highlight that the core strength of GRPO lies in filtering low-quality (especially uniformly incorrect ones) samples, rather than in its explicit reward normalization. We refer to the variant that removes both correct and incorrect samples, "REINFORCE + Remove both", as **REINFORCE-Rej**, which serves as a simplified yet competitive baseline for reward-based policy optimization in LLMs.

## 4 Related Works

**Data filtering in LLM Post-Training.** Several recent works in RLHF and preference optimization explore data filtering strategies to improve training quality. For example, Yuan et al. (2024); Dong et al. (2024); Xiong et al. (2024); Shen & Zhang (2024) discard the candidates except for the top and bottom-ranked responses to reduce noise in pairwise comparisons during RLHF learning. Yu et al. (2025a) further incorporates reward and length information of rejected responses into the filtering process. For reasoning tasks, it is also common to remove prompts that are too easy or too hard (Yang et al., 2024; Zhao et al., 2024), though this is typically done once before training. In contrast, our proposed REINFORCE-Rej performs filtering online throughout training. Furthermore, our study reveals a connection between the strong empirical performance of GRPO and its *implicit* data filtering mechanism. Building on this insight, the proposed REINFORCE-Rej makes this principle explicit. We hope our findings will inspire further research into simple, scalable filtering methods that can stabilize and enhance RL for LLMs without adding burdensome complexity.

**LLM for Mathematical Reasoning.** LLMs designed for (mathematical) reasoning have received significant attention, especially following the release of GPT-o1 by OpenAI (Jaech et al., 2024) and DeepSeek-R1 by DeepSeek (DeepSeek-AI et al., 2025). Earlier efforts primarily focused on building synthetic datasets and applying supervised fine-tuning (Gou et al., 2023; Yue et al., 2023; Yu et al., 2023; Toshniwal et al., 2024). In contrast, these new models (o1 and R1) adopt RL with verifier-based rewards as their main training approach. A key difference is that models like GPT-o1 and DeepSeek-R1 use more complex reasoning strategies—such as backward search and self-correction—and tend to generate longer outputs at inference time for better performance. Their success has inspired a surge of open-source efforts to replicate or adapt these training strategies to other domains (Jin et al., 2025; Xiong et al., 2025; Wang et al., 2024). Notably, GRPO has become the default RL method in many of these projects, often without justification. However, whether GRPO is truly better than REINFORCE, and (if the answer to the first question is yes) what contributes to its performance gains, remains largely under-explored.

## 5 Conclusion

In this work, we decomposed the popular Group Relative Policy Optimization (GRPO) algorithm to identify the essential drivers of performance for REINFORCE-style methods in LLM fine-tuning. First and foremost, we find that the **online interaction loop—iteratively generating data with an improving policy—is the most critical factor for success.** A simple online method like RAFT++ dramatically outperforms its offline counterpart and is surprisingly competitive to the more complex RL methods. This insight also aligns with recent findings on the importance of online data for preference-based methods like DPO (Xiong et al., 2023).

Our second key finding is the crucial role of sustained exploration. While online learning with positive-only samples yields rapid initial gains, it leads to a quick collapse in policy entropy, limiting the model's final capabilities. We demonstrated that one primary benefit of incorporating negative signals, as GRPO does, is

to **maintain this policy diversity, which is essential for long-term improvement** and achieving the best possible final performance.

Finally, we pinpointed the mechanism behind GRPO's stability and effectiveness. It is not its complex reward normalization, but rather its **implicit data filtering** of "extreme" prompts where all responses are uniformly correct or incorrect. Based on this insight, we introduced REINFORCE-Rej, a minimal algorithm that applies this filtering principle directly to vanilla REINFORCE. It matches or exceeds GRPO's performance with greater stability, proving that strategical data filtering is a powerful and efficient technique.

Looking forward, we believe that principled data filtering is a highly promising and underexplored avenue for designing better RL algorithms for LLMs. Our work utilizes a simple filter based on the empirical pass rate of a prompt, but we envision future methods could leverage more sophisticated criteria—such as sample uncertainty (perplexity), prompt difficulty, or reward variance—to further stabilize and accelerate training.

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

# A Additional Results

## A.1 Full Table

| Model | Algorithm | Math500 | Minerva Math | Olympiad Bench | Average |
|-------|-----------|---------|--------------|----------------|---------|
| Qwen2.5-Math-7B-base | Base | 41.3 | 11.0 | 18.6 | 23.6 |
| | RAFT | 77.4 | 40.8 | 38.6 | 52.3 |
| | RAFT++ | 80.2 | 44.9 | 43.3 | 56.1 |
| | Iterative DPO | 76.0 | 31.2 | 39.3 | 48.8 |
| | REINFORCE | 80.1 | 40.7 | 40.9 | 53.9 |
| | GRPO | 81.3 | 45.5 | 42.2 | 56.3 |
| | REINFORCE-Rej | 81.9 | 44.2 | 43.1 | 56.4 |
| LLaMA-3.2-3B-instruct | Base | 26.3 | 7.4 | 5.5 | 13.1 |
| | RAFT | 46.1 | 17.6 | 13.9 | 25.9 |
| | RAFT++ | 47.4 | 19.1 | 16.3 | 27.6 |
| | REINFORCE | 45.9 | 13.7 | 13.0 | 24.2 |
| | GRPO | 49.2 | 19.3 | 16.8 | 28.4 |
| | REINFORCE-Rej | 50.1 | 19.3 | 16.1 | 28.5 |

Table 4: Performance of different algorithms across three benchmarks including Math500 (Hendrycks et al., 2021), Minerva Math (Lewkowycz et al., 2022), and Olympiad Bench (He et al., 2024). The reported accuracy is average@16 with a temperature 1.0 and a maximal generation length of 4096 tokens.

For completeness, we present the full table of the model evaluation results in Table 4, where we also include Iterative DPO as an additional baselines.

The standard DPO algorithm optimizes a contrastive loss using a dataset of preference pairs $\{(x, a^+, a^-)\}$, where response $a^+$ is preferred over $a^-$:

$$\mathcal{L}^{\mathrm{DPO}}(\theta) = -\log \sigma\Big(\beta \log \frac{\pi_\theta(a^+|x)}{\pi_{\mathrm{ref}}(a^+|x)} - \beta \log \frac{\pi_\theta(a^-|x)}{\pi_{\mathrm{ref}}(a^-|x)}\Big),$$

where $\pi_{\mathrm{ref}}$ is a fixed reference policy. A key distinction in modern implementations is how this preference data is collected. While the original DPO algorithm was trained on a static, offline dataset, subsequent work has shown that performance is significantly improved by using an iterative scheme (Liu et al., 2023; Xiong et al., 2023; Xu et al., 2023; Hoang Tran, 2024; Dong et al., 2024). Mirroring our findings on the importance of online data, these Iterative DPO methods periodically use the latest model checkpoint to generate new responses, which are then labeled to create fresh, on-policy preference data for subsequent training rounds.

## A.2 The Impact of PPO-style Enhancements

**Importance Sampling and Clipping Improve Sample Efficiency and Performance.** The RAFT++ baseline builds upon vanilla RAFT by incorporating two key enhancements from PPO: importance sampling (for distribution correction) and objective clipping. These techniques are designed to improve sample efficiency by allowing for multiple gradient updates on a single batch of on-policy data. As shown in our main results (Table 4) and further illustrated in Figure 5, these additions are effective. The learning dynamics show that while both methods steadily improve performance, RAFT++ converges faster and achieves a higher final accuracy than the simpler vanilla RAFT.

**Clipping is Crucial for Training Stability.** To isolate the contribution of each component, we also evaluated an intermediate variant that uses importance sampling without the PPO-style clipping. Surprisingly, as shown in Figure 3, this variant underperformed even vanilla RAFT. This finding appears to contradict Ahmadian et al. (2024), who suggest that clipping is often unnecessary because it is rarely activated. We hypothesize that while clipping events may be infrequent, they are critical for stability. These events occur precisely when the updated policy, $\pi_\theta$, significantly deviates from the data-generating policy, $\pi_{\theta_{\mathrm{old}}}$. In

such cases, the objective clipping prevents excessively large, high-variance updates that would otherwise destabilize the training process and lead to performance degradation.

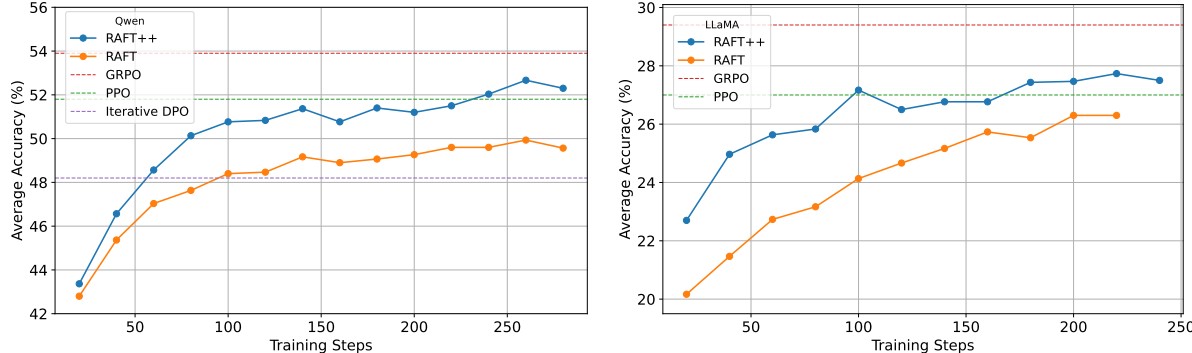

Figure 5: The learning dynamics of RAFT and RAFT++, initialized from Qwen2.5-Math-7B-base (left) and LLaMA-3.2-3B-instruct (right). The y-axis is the average@16 accuracy, that is further averaged on MATH500, Minerva Math, and Olympiad Bench. We also plot the best model of GRPO, PPO, and Iterative DPO for reference.

