# OpenReview forum: "A Minimalist Approach to LLM Reasoning: from Rejection Sampling to Reinforce"
_TMLR — Withdrawn by Authors_

### Review · Reviewer_UvwQ · 2025-11-22

**Summary Of Contributions:**

Summary:
The authors aim to investigate which part of the GRPO algorithm makes it effective.
To do so, they compare it in controlled tests vs two other RL algorithms as baselines - RAFT (fine-tuning on positive-reward samples only) and REINFORCE.
They find that the main driver of performance in GRPO stems from the implicit filtering of prompts in which all responses get the same-sign reward (all positive or all negative, which leads to a zero gradient).
They thus suggest a variant of REINFORCE with this filtering done explicitly, which matches GRPO's performance.

While the goal of the paper is very interesting and most experimental details are clear, there are still quite a few confounders and missing information in writing that raise quite a few issues. Please see below for expansion.

**Additional Comments:**

I have quite a few comments on writing and unclear experimental settings:

1. The abstract is a bit convoluted. For example, in the 3 hierarchy points, (I) and (iii) both present a main cause for GRPO success ((i) discusses "a dominant driver for performance", (iii) discusses "GRPOs main benefit"). A more ordered hierarchy would be better, indicating which of the two is the more important (and not the current ambiguous phrasing).

2. The above comment is true also for the introduction (see next comment) and to some extent the conclusion. A more ordered framing is needed in all three sections in my opinion.

3. In contribution (1) in the introduction, titled "Online exploration is the desicive factor", GRPO is contrasted with RAFT (also an on-policy algorithm). Thus, the title is unrelated to the content of the paragraph, and should probably be changed.

4. The definition of b(x) in section 2.2 is missing (introduced only later, in 2.3).

5. The equations at section 2.3 are slightly confusing. The authors present A_GRPO but don't explicitly state how it combines into the reward/loss calculations of GRPO until section 2.4 (in equation 4). Please consider re-ordering the presentation of this section for better readability.

6. In the "hyper-parameters" paragraph at section 3.1, the authors mention a baseline of iterative DPO. This baseline isn't mentioned before, when the other methods are presented, and thus feels out-of-place. I would add a presentation of DPO as a baseline to section 2 (move from appendix A.1.) OR move this mention (and any other mentions of DPO) to the appendix (since results for iterative DPO only appear in the appendix).

7. In the "evaluation" paragraph at section 3.1, the authors state "We do not include the popular AIME2024 benchmark" - the authors also don't include many other popular benchmarks. This is compltely fine, and IMO no need to mention any benchmarks not included (maybe move this to an appendix). Instead, perhaps mention the total number of prompts evaluated from each benchmark that IS included (ok to put this in appendix).

8. In section 3.2, the authors write "to disentangle these factors" - which factors?

9. How does the offline RS baseline work? What is the algorithm? Except for using offline sampling, this is unclear. I would avoid presenting an additional baseline in the middle of the results section, and move this presentation to the algorithms section.

10. Why are the results for Llama-3.2 + Offline-RS missing?

11. In non-negative GRPO (yet another algorithm variant presented in the results section), are the gradients masked when the reward for a prompt-response is negative? Or when the advantage of a specific prompt-response is negative? Unclear.

12. Why are the results in Fig4 missing for Qwen? This should appear in the appendix.

**Audience:**

Yes

**Audience Explanation:**

Any findings that help better understand WHY specific algorithms work well, additionally to presenting good ablations on these algorithms, are key to improving them further, and are of high interest to the ML community.

**Broader Impact Concerns:**

None.

**Claims And Evidence:**

No

**Claims Explanation:**

Some of the claims made are supported by good ablation experiments. However, the presentation is very messy and it isn't 100% clear that the evidence fully support the made claims without any confounders.

To expand on this:

1. Lack of statistical significance results (multiple seeds, std bars for accuracy results) makes the results less reliable.

2. While I understand the authors using the same hyperparameters across methods to compare apples to apples, it might lead to some incorrect comparisons. An *additional* comparison of the ideal hyperparameters for each method separately will provide stronger evidence. This isn't mandatory though.

3. Since the authors don't explain fully what is offline-RS (see in "Additional comments") it is unclear if this is a proper control experiment that validates the importance of the online nature of GRPO (which is the first and main claim of the authors). Also, the missing results for Offline-RS for Llama raise suspicion.

4. In section 3.2., the authors compare RAFT++ to GRPO. These two methods differ by two things: GRPO incorporates gradients from negative rewards while RAFT++ doesn't, AND GRPO uses reward shaping (mean and std or reward to find the advantage). The authors don't do an experiment to separate between these two confounders in figure 2, and thus it is unclear (in the context of section 3.2.) that the claim of "negative signals are valuable for sustaining exploration" is correct.

5. In section 3.3, the authors compare REINFORCE (and their variants of it) to GRPO. These two methods differ (according to the authors, bottom of page 7) in "the way of introducing negative samples and the application of reward normalization". Here as well, the control experiments don't exactly capture these two confounders (see comment). It's unclear how checking "REINFORCE+remove all correct" is neccesary. Second, there is no experiment that fully incorporates reward normalization to REINFORCE (only (2) incorporates mean subtraction, but no std reduction).

**Requested Changes:**

1. Critical: I request the authors highlight that their findings apply to the binary-reward settings, and not to general RL.

2. Critical: Main requested change - I would ask the authors to clearly explain how their control experiments in section 3.2 and 3.3 capture the confounders between GRPO and REINFORCE and RAFT++, or perform control experiments that properly differentiate between these confounders (See further above, in "Are the claims made supported by evidence").

3. Critical: There are missing results - Fig4 for Qwen, Offline-RS results for Llama3.2. Please add these results to the main paper and the appendix.

4. While the experiments are mostly straight-forward and clear, there are many points in the writing that can be improved. Please see changes under "Additional comments" and fix them for a clearer revision.

5. I would ask that the authors re-run the experiments for multiple train/test splits (if computationally feasible), and present the resulting variation results (in a form of error bars in accuracy graphs). If this isn't feasible, I expect some explanation on why.

---

### Review · Reviewer_Z1oF · 2025-11-28

**Summary Of Contributions:**

This paper investigates and breaks down the effectiveness of GRPO for reasoning workloads in LLM/LRM training. By benchmarking GRPO against simpler baselines, the authors uncover the following insights:
- The crucial driver of performance in reasoning capabilities is the online data collection loop
- Negative signals are crucial for sustaining exploration and preventing entropy collapse
- GRPO's core benefit comes from the implicit data filtering effect (i.e., removing prompts where responses are all-correct or all-wrong)
- Based on the insight, the authors proposed REINFORCE-REJ, a simple method that extends REINFORCE by explicitly filtering out select prompts. The proposed method achieves performance on par with GRPO.

Key strengths:
- Rigorous and systematic ablation study that entangles the components of GRPO
- Novel insight: GRPO's effectiveness mainly comes from filtering "extreme" prompts
- Simplicity of REINFORCE-REJ

Key weaknesses:
- No significant weaknesses except for a lack of qualitative analysis of the destabilizing prompts and the limited workloads.

**Audience:**

Yes

**Audience Explanation:**

- GRPO is a trendy approach for LLM training, and in particular, LRMs that drive inference-time scaling
- The writeup of the background (connecting RAFT, REINFORCE, and GRPO) is clean, easy to understand, and educational for non-experts
- The insight into GPRO's data filtering effect is novel and interesting
- The proposed REINFORCE-REJ is intuitive, simple, and effective

**Broader Impact Concerns:**

No broader impact concerns.

**Claims And Evidence:**

Yes

**Claims Explanation:**

- The construction of baselines in §3 is reasonable, and the experiment methodology is sound. I am very fond of how the authors conducted experiments to unearth insights!
- The authors' contribution, REINFORCE-REJ, achieves performance close to GRPO

**Requested Changes:**

- Add a brief, qualitative analysis of example prompts for which the responses are either all-correct or all-wrong. These types of questions should be filtered, and seeing concrete examples would help clarify why they are destabilizing. Are the all-wrong prompts simply extremely difficult for the model to answer, or are they ambiguous? E.g., could it be that the correct answer and the LLM's output are semantically equivalent ("\bbox{7/3}" vs. "\bbox{\frac{7}{3}}"), but the verifier is a simple exact-match metric?
- A limitation of the paper is that it only includes math reasoning workloads, so a potential improvement is to evaluate the method on more non-math reasoning workloads, e.g., GPQA. However, I acknowledge that it is not a simple addition, so this is a good-to-have that could strengthen the work, not a must-have.

---

### Review · Reviewer_X9Xk · 2025-11-28

**Summary Of Contributions:**

The paper aims to dissect GRPO to identify the underlying factors behind its strong empirical performance. The authors’ findings can be summarized as follows:
1. Iterative data collection plays a significant role in the method’s effectiveness, and
2. Data quality is critical, with the experiments showing that rejecting all incorrect or all correct samples has a larger impact on performance than the group-wise normalization used in GRPO.

**Audience:**

Yes

**Audience Explanation:**

I believe there remain many unknowns regarding the success of RLHF, in large part due to limited experimental rigor in prior work. A paper that systematically evaluates and tests these claims is therefore valuable to the community.

**Broader Impact Concerns:**

No ethical concern.

**Claims And Evidence:**

No

**Claims Explanation:**

Overall, the study is interesting, but I will focus on the aspects that require clarification to strengthen the contribution. In my view, the work still lacks rigor in several places, and I detail the points that remain unclear in the Requested Changes section.

**Requested Changes:**

The paper lacks a comparison and thorough discussion of the recent work by Ahmadian et al., who show that a simple REINFORCE-style or vanilla policy gradient method can perform outperform RAFT and PPO. This discussion is highly relevant to the core questions the authors are investigating and should appear in the introduction. The approach should also be compared against these baselines in the experiments. Is data rejection an important issue in that paper too?

Additionally, note that [1] uses two different choices of baselines in the REINFORCE objective for estimating the advantage, and they argue that clipping or modeling partial completion is unnecessary. In contrast, Sections 2.3 and 2.4 of this paper suggest that even the REINFORCE model uses the PPO-style objective in Eq. 4, which includes clipping and a sampling ratio. If this interpretation is correct, the method should not be described as REINFORCE, as it is effectively a variant of PPO that uses an estimated advantage and partial completion. The paper should clarify this point.

The claim that GRPO outperforms RAFT++ in Fig. 1 is not well supported. The results must be run over multiple seeds with confidence intervals. Even visually, the curves appear too close to conclude a meaningful difference. The results in Fig. 2 are more suggestive, but they also require multiple seeds and confidence intervals to be conclusive.

There is also no discussion of why REINFORCE + Mean Zero performs so well, despite not using data rejection. Isn’t this the simplest thing to do? Understanding this behavior seems important for the paper’s central claims.

Finally, the authors should not refer to “REINFORCE-Rej” as their own model. Dissecting existing methods by disabling components is valuable, but presenting a minor ablation of a prior method as a new model is misleading.

[1] Ahmadian, A., Cremer, C., Gallé, M., Fadaee, M., Kreutzer, J., Pietquin, O., Üstün, A. and Hooker, S., 2024. Back to basics: Revisiting reinforce style optimization for learning from human feedback in llms. arXiv preprint arXiv:2402.14740.

---

### Comment · Action_Editor_Jj5H · 2025-11-29
**Discussions between authors and reviewers**

Dear authors,

Please read through the reviews, and any requested changes. The discussion period lasts two weeks. I kindly ask that you start your responses as soon as possible to help facilitate a constructive discussion.

---

### Note · Authors · 2025-12-09

I have read and agree with the venue's withdrawal policy on behalf of myself and my co-authors.